# Elasticity-controlled jamming criticality in soft composite solids

Yiqiu Zhao ®[1] ✉, Haitao Hu[1], Yulu Huang[1], Hanqing Liu ®[2], Caishan Yan ®[1], Chang Xu ®[1], Rui Zhang ®[1], Yifan Wang ®[3] & Qin Xu ®[1] ✉

Soft composite solids are made of inclusions dispersed within soft matrices. They are ubiquitous in nature and form the basis of many biological tissues. In the field of materials science, synthetic soft composites are promising candidates for building various engineering devices due to their highly programmable features. However, when the volume fraction of the inclusions increases, predicting the mechanical properties of these materials poses a significant challenge for the classical theories of composite mechanics. The difficulty arises from the inherently disordered, multi-scale interactions between the inclusions and the matrix. To address this challenge, we systematically investigated the mechanics of densely filled soft elastomers containing stiff microspheres. We experimentally demonstrate how the strain-stiffening response of the soft composites is governed by the critical scalings in the vicinity of a shear-jamming transition of the included particles. The proposed criticality framework quantitatively connects the overall mechanics of a soft composite with the elasticity of the matrix and the particles, and captures the diverse mechanical responses observed across a wide range of material parameters. The findings uncover a novel design paradigm of composite mechanics that relies on engineering the jamming properties of the embedded inclusions.

Dispersing nano-to-micron-sized particles within a soft polymeric gel forms soft composite solids that are widely used in various engineering materials, including synthetic tissue[1], wearable biomedical devices[2,3], and soft robots[4]. In addition to reinforcing the polymer matrix[5], the dispersed particles can enable diverse functional features such as anisotropic elasticity[6], shape-memory effects[7,8], and stimuli-responsive behaviors[9,10]. Due to the great compliance of soft polymeric gels, the embedded particles can undergo moderate displacement within the matrix without causing internal fractures[9]. This particle rearrangement may alter both the strain couplings among neighboring inclusions[11] and the stress fields over a large length scale[9,12]. Compared with classical stiff composite materials[13], the current understanding of the multi-scale interactions within soft composites remains very limited.

The complexity of composite mechanics increases exponentially with the volume fraction of the inclusions. In a dilute composite, the mechanics are solely determined by the interactions between an isolated inclusion and the surrounding matrix, which allows the effective shear modulus to be described by the classical Eshelby theory[14]. Further, modified effective medium theories have been extended to systems with finite-density inclusions, where neighboring particles interact via their induced strain fields[11,15]. However, this assumption of matrix-mediated, short-range interactions breaks down in the dense limit, where the overall stress response may involve networks of direct contacts[16,17] or long-range rearrangements of dispersed particles[12]. Due to the inherently disordered and heterogeneous microstructures of dense soft composites, predicting their mechanics is challenging for classical composite theories.

To address these issues, we systematically investigated the strain stiffening of soft elastomers containing a high volume fraction of stiff microspheres. Inspired by the concepts of both granular jamming[18–21]

[1]Department of Physics, The Hong Kong University of Science and Technology, Hong Kong SAR, China. [2]Theoretical Division, Los Alamos National Laboratory, Los Alamos, NM 87545, USA. [3]School of Mechanical and Aerospace Engineering, Nanyang Technological University, Singapore 639798, Singapore. ✉e-mail: yiqiuzhao@ust.hk; qinxu@ust.hk

and rigidity transitions in disordered systems[22–24], we demonstrate that the mechanical responses of soft composites are governed by elasticity-controlled scalings near a continuous phase transition. In the absence of matrix elasticity, the transition coincides with the shear-jamming of the included particles. This novel criticality framework captures the stiffening responses for a variety of material parameters where the classical theories break down. The results provide a new approach to understand the nonlinear mechanical responses of various multi-phase soft materials.

## Results

### Strain-stiffening responses of soft composite solids

We prepared compliant polydimethylsiloxane (PDMS) elastomers filled with stiff polystyrene (PS) microspheres having an average diameter of 30 μm (Fig. 1a and Supplementary Fig. 8). While the shear modulus of the PS spheres is $G_p = 1.6$ GPa (Supplementary Fig. 1), the shear modulus of the PDMS matrix was systematically varied from $G_m = 0.04$ to 4 kPa by tuning the crosslinking density (ref. 25 and Supplementary Fig. 7). The mechanical properties of the soft composites were characterized using a rheometer equipped with a parallel-plate shear cell (Fig. 1b). The top plate controls the gap size ($d$) and applies axial compressive strains ($\varepsilon$) (Supplementary Fig. 9). Due to the incompressibility of crosslinked PDMS gels[26], the volume of the sample remains unchanged under axial compression (Supplementary Fig. 12 and Supplementary Movie 1), which gives rise to pure shear. At each given $\varepsilon$, the rheological properties of the composites were measured using an oscillatory shear with a small amplitude ($\delta\gamma_a = 0.01\%$). At an angular frequency ($\omega = 0.1$ rad/s), the storage modulus has reached a low-frequency plateau (Supplementary Fig. 11), which indicates the shear modulus of soft composites ($G$). This resulting $G(\varepsilon)$ represents the linear elastic response of the soft composites in differently sheared states (Supplementary Fig. 10).

The dense soft composites exhibit characteristic strain-stiffening responses under the axial compressions (Fig. 1c). The stiffening degree is determined by both the particle volume fraction $\phi$ and the shear modulus of the elastomer matrix $G_m$. First, at a fixed $G_m = 1.28$ kPa, the relative shear modulus, $G_r = G/G_m$, grows more rapidly with $\varepsilon$ as $\phi$ increases from 0.44 to 0.67. Second, at a fixed $\phi = 0.60$, the strain stiffening becomes more pronounced while $G_m$ decreases from 1.28 to 0.04 kPa.

We define $G_{r,max}$ as the relative shear modulus at the maximally stiffened states and $G_{r,0}$ as the relative shear modulus at $\varepsilon = 0$. Within experimental uncertainty, $G_{r,max}$ appears at approximately $\varepsilon = 0.2$ regardless of $\phi$ and $G_m$. Therefore, we estimated $G_{r,max}$ for all the samples using the values of $G_r$ at $\varepsilon = 0.2$. For $\varepsilon > 0.2$, $G_r$ decreases with $\varepsilon$, and the composites were unable to fully recover their original shapes after the compressions were removed. This plasticity is likely caused by internal fractures between the elastomer and the particles[27]. Since the adhesion energy at gel interfaces is approximately independent of the crosslinking density[28], the plasticity onset ($\varepsilon \approx 0.2$) remains nearly unchanged for various $G_m$. In contrast, the plots of $G_r(\varepsilon)$ appear to be highly reproducible when the compressions are released at $\varepsilon < 0.2$. In this study, we focus exclusively on the stiffening regime between $\varepsilon = 0$ and 0.2.

Figure 1d shows both $G_{r,max}$ (solid points) and $G_{r,0}$ (hollow points) as a function of $\phi$ as $G_m$ varies between 0.04 and 1.28 kPa. For $\phi < 0.4$, only $G_{r,0}$ was reported since no strain-stiffening was found. For comparison with the classical theories of composite mechanics, we plotted the predictions from the Eshelby theory[14] and the Mori–Tanaka approximation scheme[15], which align well with the $G_{r,0}$ measured in the dilute limit ($\phi < 0.2$). However, for dense composites ($\phi > 0.4$), the classical theories significantly deviate from the measured $G_{r,max}$ and $G_{r,0}$, and also fail to describe the strain-dependent shear modulus $G_r(\varepsilon)$. These mismatches suggest that potential mechanisms, such as direct contact between inclusions[16,17], were overlooked in the classical models of the mechanics of dense soft composites.

### Signatures of jamming-controlled elasticity

We re-examine the super-exponential rise of $G_{r,max}$ in Fig. 1d. As $G_m$ decreases, the growth of $G_{r,max}$ becomes increasingly more divergent

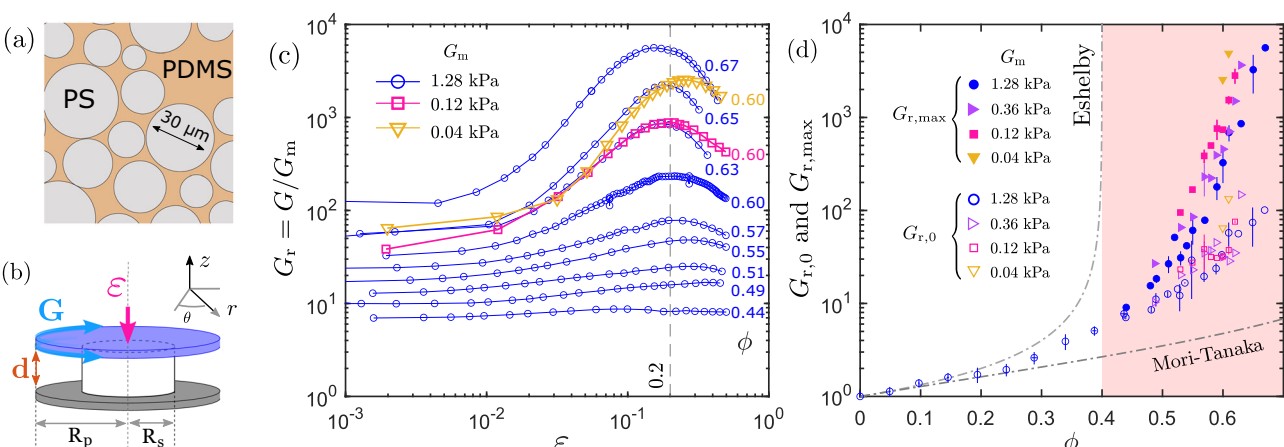

**Fig. 1 | Strain-stiffening of PS-PDMS soft composites under volume-conserving compressions. a** Schematic of the cross-section of PS-PDMS composites. For a predetermined volume fraction $\phi$, polydisperse PS spheres with an average diameter of approximately 30 μm are well dispersed in a crosslinked PDMS matrix. **b** Schematic of the experimental setup used to characterize the strain-stiffening of the soft composites. The top plate moves down in a stepwise manner to apply an axial strain $\varepsilon$. At each $\varepsilon$, the linear shear modulus $G$ was measured through an oscillatory shear with a strain amplitude of $\delta\gamma_a = 10^{-4}$ and an angular frequency of $\omega = 0.1$ rad/s. **c** Plots of the relative shear modulus, $G_r = G/G_m$, against $\varepsilon$ for various particle volume fractions $\phi$ and matrix shear moduli $G_m$. The blue hollow circles indicate the results for a constant $G_m = 1.28$ kPa as $\phi$ increases from 0.44 to 0.67.

In addition, the hollow red squares and hollow yellow triangles represent the results of $G_r(\varepsilon)$ at the same $\phi = 0.60$ but for different matrix moduli, $G_m = 0.12$ kPa and $G_m = 0.04$ kPa, respectively. **d** Comparison between the experimentally measured $G_r$ and the predictions from the classical theories of composite mechanics. The solid and hollow points indicate $G_{r,max}$ and $G_{r,0}$, respectively, versus $\phi$ for samples with varying $G_m$. The error bars represent the standard deviation from measuring two to five independently fabricated samples. The two dashed gray lines represent the predictions from the Eshelby theory and the Mori–Tanaka approximation. The pink area represents the range of the volume fraction where strain-stiffening was observed.

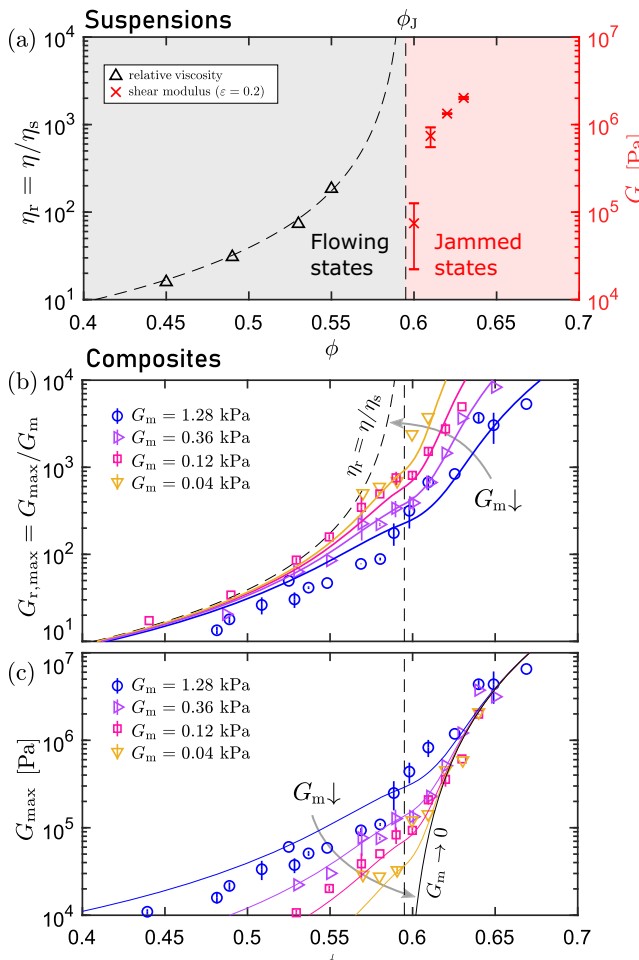

**Fig. 2 | Signatures of jamming-controlled elasticity. a** Rigidity transition of PS particles suspended in un-crosslinked silicone oil. The black triangles show the relative viscosity $\eta_r = \eta/\eta_s$ in the low-stress Newtonian regime for different particle volume fractions $\phi = 0.45, 0.49, 0.53, 0.55$. The dashed black curve indicates the best fit of the experimental results to Eq. (1) where $\phi_J = 0.594$. In the regime of $\phi > \phi_J$, the suspensions were initially unjammed at $\varepsilon = 0$, and then shear jammed at $\varepsilon = 0.2$. The red crosses represent their shear moduli ($G_s$) in the shear-jammed states ($\varepsilon = 0.2$). The error bars represent the standard deviation from five independent measurements. **b** Plots of $G_{r,\max}$ against $\phi$ for different $G_m$ values. To compare the absolute values of $G_{r,\max}$ with $\eta_r$, the fit to Eq. (1) obtained in panel **a** is also shown by the dashed gray line in the same plot. **c** The actual shear modulus $G_{\max}$ is plotted against $\phi$ for different $G_m$ values based on the results in (**b**). The solid lines in both panels **b** and **c** are predictions from the scaling model based on jamming criticality (Eqs. (4) and (5)). The error bars in **b** and **c** represent the standard deviation from measuring two to five independently fabricated samples.

near $\phi \approx 0.6$. Since a soft composite solid will asymptotically become a granular suspension as the matrix elasticity approaches zero, we hypothesize an underlying connection between the shear-jamming of dense suspensions and the strain-stiffening of soft composites in the limit of $G_m \to 0$.

To validate this assumption, we first characterize the shear rheology of a concentrated PS suspension in the PDMS base solution without any crosslinkers. We define the relative viscosity ($\eta_r$) as the ratio of the viscosity of the suspension ($\eta$) to that of the PDMS base ($\eta_s = 1.0$ Pa s): $\eta_r = \eta/\eta_s$. The left panel in Fig. 2a shows $\eta_r$ measured within a Newtonian regime where the shear stress ranges from 1 to 10 Pa. This value effectively estimates the suspension viscosity in the quasi-static limit. The details of the rheological measurements are provided in Supplementary Fig. 2. The results are well described by

the Krieger–Dougherty relation[29,30]

$$\eta_r(\phi) = \frac{\eta(\phi)}{\eta_s} = (1 - \phi/\phi_J)^{-\gamma}, \; (\phi < \phi_J) \quad (1)$$

with a fixed exponent $\gamma = 2$ and a fitted jamming volume fraction $\phi_J = 0.594 \pm 0.003$. A similar scaling has been identified in the simulations of over-damped granular systems near jamming[18,31]. For $\phi > \phi_J$, we did not observe homogeneous steady shear flow at any shear rate. Instead, the suspensions were consistently jammed under continuous shear (Supplementary Fig. 5). To quantify the mechanical responses of these shear-jammed states, we initially prepared fully relaxed suspensions without rigidity at $\varepsilon = 0$. Subsequently, we applied an axial strain $\varepsilon > 0$ to induce jamming in the suspensions and measure their shear moduli. The measurement protocol is detailed in Supplementary Information Section I.A.3. The right panel of Fig. 2a represents the nonzero shear moduli of the shear-jammed PS-PDMS suspensions ($G_s$) measured at $\varepsilon = 0.2$ in the regime of $\phi > \phi_J$. Since no significant change in $G_s$ was found when $\varepsilon$ was further increased (Supplementary Fig. 4), $\phi_J = 0.594$ represents the lowest particle volume fraction required to achieve shear-jamming in the PS-PDMS suspensions.

In Fig. 2b, we plot $\eta_r(\phi)$ from Eq. (1) together with $G_{r,\max}(\phi)$ of the composites for a comparison. The traces of $G_{r,\max}$ gradually converge to $\eta_r$ as $G_m$ decreases, suggesting that $G_{r,\max} \approx (1 - \phi/\phi_J)^{-\gamma}$ for $\phi < \phi_J$ as $G_m$ approaches zero, and the actual shear modulus $G_{\max}$ scales linearly with $G_m$ in this limit. In contrast, for $\phi > \phi_J$, $G_{\max}$ becomes independent of $G_m$ (Fig. 2c) and is close to the value of $G_s$ measured independently from the jammed suspensions (Fig. 2a), suggesting a particle-dominated response. Considering the contrasting mechanical behaviors exhibited for the ranges $\phi < \phi_J$ and $\phi > \phi_J$, it is likely that the shear-jamming point of the suspensions controls a crossover of the mechanical properties of the composites.

### Elasticity-controlled criticality near jamming

Since the plots of $G_{\max}(\phi)$ in Fig. 2c resemble the critical behaviors near a continuous phase transition[23,24,32,33], we next investigate the scalings of the composite shear modulus ($G_{\max}$) near ($\phi = \phi_J$, $G_m = 0$). Motivated by the observation that $G_{r,\max}$ approaches $\eta_r$ as $G_m \to 0$ (Fig. 2b) and the classical analogy between the effective shear modulus and the shear viscosity in multi-phase systems[34–36], we conjecture the scaling law

$$\lim_{G_m \to 0} \frac{G_{\max}(\phi)}{G_m} = \frac{\eta(\phi)}{\eta_s} = (1 - \phi/\phi_J)^{-\gamma}, \; (\phi < \phi_J) \quad (2)$$

with $\gamma = 2$ and $\phi_J = 0.594$. To demonstrate the validity of this scaling assumption, we plot $G_{r,\max}$ against $\phi_J - \phi$ in Fig. 3a with different $G_m$ values, where the results show the best agreement with Eq. (2) for the softest matrix. We further consider how $G_{\max}$ varies with $G_m$ at $\phi = \phi_J$. In Fig. 3b, $G_{\max}$ is plotted at $\phi = 0.59 \approx \phi_J$ against $G_m$, which can be fitted to the power-law scaling

$$G_{\max} \sim G_m^{1/\delta}, \; (\phi = \phi_J) \quad (3)$$

with a fitted exponent $1/\delta = 0.6 \pm 0.1$.

Considering the scalings in Eqs. (2) and (3), we compare the soft composites near $\phi_J$ with a ferromagnetic system near the Curie temperature ($T_c$). The material parameters of the soft composites ($G_{\max}, G_m, \phi - \phi_J$) are directly analogous to ($M, H, T - T_c$) in the Ising model. By assuming a scale-invariant free energy at the critical point ($\phi = \phi_J$, $G_m = 0$), we propose a universal scaling form

$$G_{\max} = |1 - \phi/\phi_J|^\beta f_\pm \left( \frac{G_m}{|1 - \phi/\phi_J|^\Delta} \right) \quad (4)$$

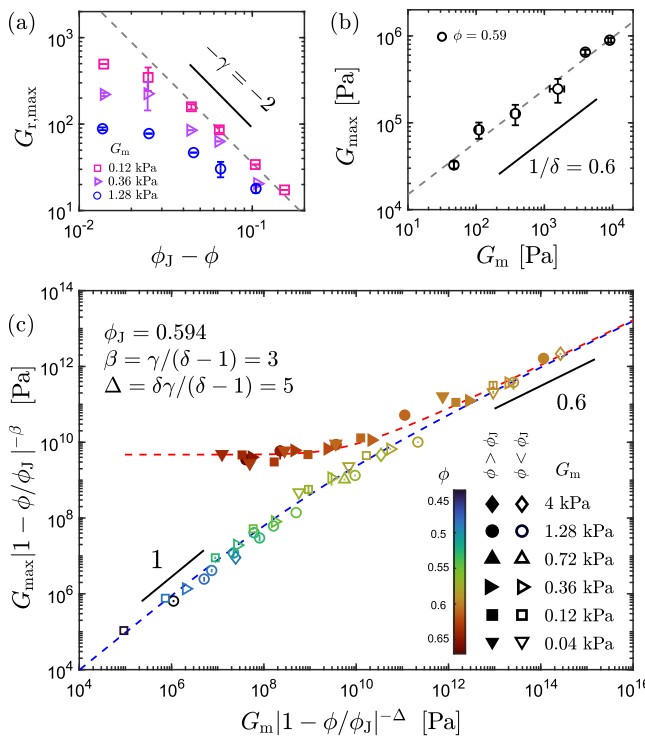

**Fig. 3 | Elasticity-controlled criticality near jamming. a** Plots of $G_{r,max}$ against $\phi_J - \phi$ for $G_m = 0.12$, 0.36, and 1.28 kPa, respectively. The dashed gray line indicates the scaling law of Eq. (2). **b** Plots of $G_{max}$ versus $G_m$ for composites with $\phi = 0.59 \approx \phi_J$, where the dashed black line represents the scaling law of Eq. (3). **c** Scaling collapse of $G_{max}$, normalized by $|1 - \phi/\phi_J|^{\beta}$, as a function of $G_m/|1 - \phi/\phi_J|^{\Delta}$ with $\phi_J = 0.594$, $\beta = 3$, and $\Delta = 5$. The solid markers represent the experimental results obtained for $\phi > \phi_J$, and the open markers represent the results obtained for $\phi < \phi_J$. The data points are labeled with different colors based on $\phi$. The dashed red and blue curves are the best fits to the equations of state (Eq. (5)) for the experimental results within $\phi > \phi_J$ and $\phi < \phi_J$, respectively. All error bars represent standard deviations from measuring two to five independently fabricated samples.

where $\beta = \gamma/(\delta - 1) = 3.0 \pm 0.7$ and $\Delta = \delta\beta = 5.0 \pm 1.1$, and the crossover scaling functions $f_+$ and $f_-$ apply to the regimes of $\phi > \phi_J$ and $\phi < \phi_J$, respectively. The derivations of Eq. (4) and the relationships between the exponents are described in the "Methods" section. A similar scaling was previously applied to study fibrous networks near central force rigidity transitions, where the bending rigidity plays a similar role as $G_m$ in soft composites[23,33].

To test our scaling ansatz (Eq. (4)), the mechanical responses ($G_{max}$) measured for different $G_m$ and $\phi$ are plotted in Fig. 3c using the rescaled variables $G_{max}/|1 - \phi/\phi_J|^{\beta}$ and $G_m/|1 - \phi/\phi_J|^{\Delta}$. The range of $G_m$ spans two orders of magnitude, from 0.04 to 4.0 kPa, while $\phi$ increases from 0.45 to 0.67 around $\phi_J = 0.594$. Consistent with Eq. (4), the data points for $\phi > \phi_J$ and $\phi < \phi_J$ are nicely collapsed onto two distinct branches. The $\phi < \phi_J$ branch exhibits a slope close to 1, indicating that $G_{max} \sim G_m$. The $\phi > \phi_J$ branch reaches a plateau independent of $G_m$, suggesting that $G_{max}$ is dominated by the particle phase. In the limit of $\phi \approx \phi_J$, a critical regime emerges where the two branches become indistinguishable and both follow the same scaling, $G_{max} \sim G_m^{\beta/\Delta} \sim G_m^{0.6}$.

To model $G_{max}$ analytically, we derived an explicit form of the equations of state

$$\tilde{h} = g_\pm(\tilde{m}) = c_1 \tilde{m}^{\Delta/\beta} \mp c_2 \tilde{m}^{(\Delta-1)/\beta} \mp \tilde{m} \tag{5}$$

where $g_\pm$ are the inverse functions of $f_\pm$. The reduced variables $\tilde{h} \equiv G_m G_p^{-1}|1 - \phi/\phi_J|^{-\Delta}$ and $\tilde{m} \equiv G_{max} G_p^{-1}|1 - \phi/\phi_J|^{-\beta}$ were used to simplify the notation. The derivation is detailed in the "Methods" section. By fitting the data in Fig. 3c to Eq. (5), we obtain the

material constants $c_1 = 1.4$ and $c_2 = 1.3$ for the PS–PDMS composites. With all the essential parameters ($\phi_J$, $\beta$, $\Delta$, $c_1$, and $c_2$), we can calculate $G_{max}$ for a given $\phi$ and $G_m$. For instance, the colored solid lines in Fig. 2b and c represent the theoretical predictions from Eqs. (4) and (5).

## Criticality near a strain-dependent jamming transition

To describe the entire strain-stiffening regime, it is necessary to expand the scaling analysis to include the axial strains ranging from $\varepsilon = 0$ to 0.2. Since the shear-jamming point of granular materials depends on strain[37–45], we next explore an extension to our model by incorporating a strain-dependent jamming volume fraction $\phi_J(\varepsilon)$ for $0 \leq \varepsilon \leq 0.2$.

Motivated by the previous simulation showing the similar symmetry between shear-jamming and isotropic jamming transitons[40], we assume that the critical exponents ($\beta = 3$ and $\Delta = 5$) and the material parameters ($c_1 = 1.4$ and $c_2 = 1.3$) of dense composites remain constant for different $\varepsilon$. Therefore, Eq. (5) is rewritten as

$$\tilde{h}_\varepsilon = g_\pm(\tilde{m}_\varepsilon) = c_1 \tilde{m}_\varepsilon^{\Delta/\beta} \mp c_2 \tilde{m}_\varepsilon^{(\Delta-1)/\beta} \mp \tilde{m}_\varepsilon, \tag{6}$$

where $\tilde{h}_\varepsilon \equiv G_m G_p^{-1}|1 - \phi/\phi_J(\varepsilon)|^{-\Delta}$ and $\tilde{m}_\varepsilon \equiv G(\varepsilon)G_p^{-1}|1 - \phi/\phi_J(\varepsilon)|^{-\beta}$. For each $\varepsilon$, we search for an optimal $\phi_J(\varepsilon)$ that allows the composite shear modulus $G(\varepsilon)$ measured with different $G_m$ and $\phi$ to be collapsed onto Eq. (6) (the dashed gray line in Fig. 4a). As a consequence, we are able to overlay $G(\varepsilon)$ measured within the range of $0 \leq \varepsilon \leq 0.2$ by plotting $G(\varepsilon)/(G_p|1 - \phi/\phi_J(\varepsilon)|^{\beta})$ versus $G_m/(G_p|1 - \phi/\phi_J(\varepsilon)|^{\Delta})$. The resulting $\phi_J(\varepsilon)$ in Fig. 4b can be fitted to a form that describes the shear-jamming phase boundary of granular materials[39,42,46]

$$\phi_J(\varepsilon) = \phi_m + (\phi_0 - \phi_m)e^{-\varepsilon/\varepsilon^*} \tag{7}$$

with $\phi_0 = 0.688 \pm 0.004$, $\phi_m = 0.594 \pm 0.002$, and a characteristic strain scale $\varepsilon^* = 0.035 \pm 0.003$. While $\phi_m$ agrees with $\phi_J = 0.594$ measured under the steady-state rheology of the PS-PDMS suspensions shown in Fig. 2a, $\phi_0$ is consistent with the simulated random close packing of spheres having the same size distribution as our samples (Supplementary Fig. 6). Although Eq. (7) was obtained from the scaling behaviors of soft composites, it effectively predicts the line of rigidity transitions for the PS-PDMS suspensions in our experiments (Supplementary Fig. 4).

To test the universality of the scaling model, we further examined a different composite system made by dispersing glass beads in PDMS matrices. The size of these glass beads is similar to that of the PS particles but their shear modulus is ten times higher; that is, $G_p = 15.8$ GPa. The results of the glass–PDMS composites are collapsed onto the same plot in Fig. 4a with the same critical exponents $\beta = 3$ and $\Delta = 5$ but different material constants $c_1 = 0.9$ and $c_2 = 0.8$. The difference in $c_1$ and $c_2$ likely due to the high bonding energy between glass and PDMS. The resulting $\phi_J(\varepsilon)$ was also fitted to Eq. (7) with $\phi_0 = 0.676 \pm 0.003$, $\phi_m = 0.613 \pm 0.003$, and $\varepsilon^* = 0.040 \pm 0.007$. We again found that $\phi_m = 0.613$ is consistent with the shear-jamming point of the glass–PDMS suspensions and that $\phi_0 = 0.676$ is consistent with the predicted random close packing. The variation in $\phi_m$ may stem from the difference in frictional coefficients and polydispersities between the PS and glass particles.

With the given parameters $G_m$, $\phi$ and $\varepsilon$, we can calculate the shear modulus of soft composites as

$$G(\varepsilon, \phi, G_m) = G_p|1 - \phi/\phi_J(\varepsilon)|^{\beta} f_\pm\left(\frac{G_m/G_p}{|1 - \phi/\phi_J(\varepsilon)|^{\Delta}}\right), \tag{8}$$

where $\phi_J(\varepsilon)$ is given by Eq. (7), and the functions $f_\pm$ can be evaluated by numerically solving the inverse functions $g_\pm$ in Eq. (6). In Fig. 4c, we compared the measured shear moduli of two sets of PS-PDMS samples

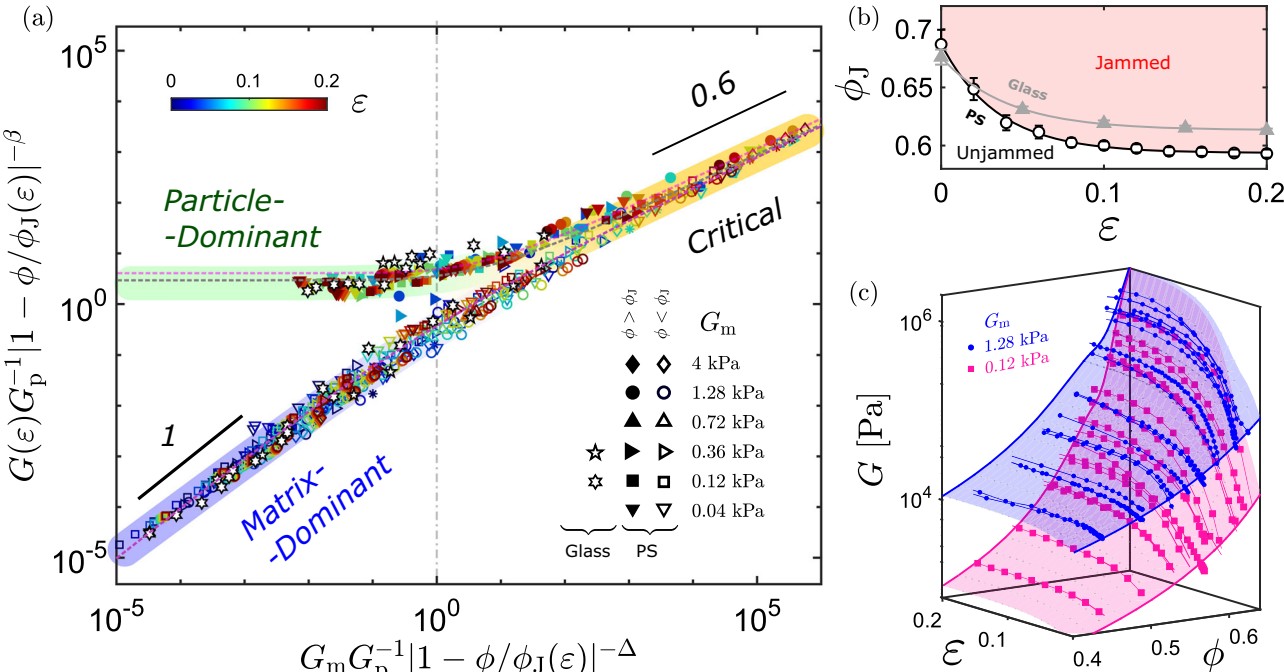

**Fig. 4 | Criticality near a strain-dependent jamming transition. a** Collapse of the rescaled composite shear modulus $G(\varepsilon)/(G_p|1-\phi/\phi_J(\varepsilon)|^\beta)$ as a function of the rescaled matrix shear modulus $G_m/(G_p|1-\phi/\phi_J(\varepsilon)|^\Delta)$ at different axial strains ($\varepsilon$) with two fixed critical exponents $\beta = 3$ and $\Delta = 5$. The data points include the experimental results obtained for two composite systems: PS-PDMS and glass-PDMS soft composites. The dotted gray (and pink) curves represent the best fit to Eq. (6) for the PS-PDMS (and glass-PDMS) composites. The vertical dashed line ($G_m/G_p = |1-\phi/\phi_J(\varepsilon)|^\Delta$) approximates the crossover boundary from the critical regime to the particle- or matrix-dominated regime. **b** Plots of the fitted $\phi_J(\varepsilon)$ for PS-PDMS

(open black circles) and glass-PDMS systems (gray uptriangles). The error bars indicate the fitting uncertainties. The solid black and gray curves represent the best fits of $\phi_J(\varepsilon)$ to Eq. (7) for these two material systems, respectively. The pink area indicates the shear-jammed phase of the PS-PDMS suspensions. **c** Plots of the shear modulus of PS-PDMS composites ($G$) as a function of both $\varepsilon$ and $\phi$. The blue and pink connected points represent the experimental results for $G_m = 1.28$ and 0.12 kPa, respectively. The blue and pink surfaces represent the theoretical predictions from Eq. (8) for these two $G_m$ values.

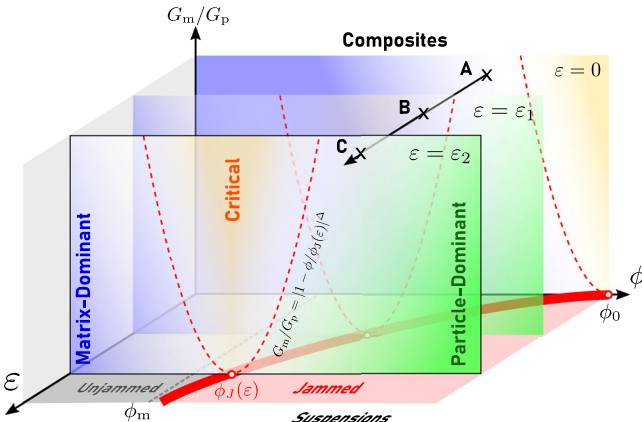

**Fig. 5 | Phase diagram of the mechanical responses of soft composite solids and granular suspensions.** The $G_m = 0$ plane represents the suspensions consisting of particles dispersing in uncrosslinked polymers. The solid red line in the $G_m = 0$ plane signifies the shear-jamming transition ($\phi_J(\varepsilon)$) of dense suspensions[37]. The 3D space defined by $G_m > 0$ represents soft composites consisting of particles dispersing in crosslinked polymeric elastomers. The mechanical properties of dense soft composites under different strains $\varepsilon$ are controlled by the scalings (Eq. (8)) near the critical line $\phi_J(\varepsilon)$. The dashed red lines $G_m/G_p = |1-\phi/\phi_J(\varepsilon)|^\Delta$ indicate the crossover boundary from the matrix- or particle-dominated regime to the critical regime. The solid arrow ($A \rightarrow B \rightarrow C$) illustrates a representative strain-stiffening process of soft composites with a particle volume fraction $\phi_m < \phi < \phi_0$. With the increase in the applied strain $\varepsilon$, the mechanical response of the composites crosses over from the matrix-dominated regime ($\varepsilon = 0$) to the critical regime ($\varepsilon = \varepsilon_1$), and finally to the particle-dominated regime ($\varepsilon = \varepsilon_2$).

with $G_m = 0.12$ and 1.28 kPa, respectively, to the theoretical predictions from Eq. (8).

The phase diagram in Fig. 5 summarizes the fundamental aspects of our criticality framework. The $G_m = 0$ plane represents the granular suspensions consisting of particles in uncrosslinked polymers. The solid red curve within the plane, $\phi = \phi_J(\varepsilon)$, denotes the boundary of the shear-jamming transition[21,37]. Soft composites exist in the 3D space characterized by $G_m > 0$, and the vertical planes in Fig. 5 represent the cross sections of this space at different strains. While there is no rigidity transition in this space with $G_m > 0$, the mechanics are determined by the critical scalings near $\phi_J(\varepsilon)$. When $G_m/G_p \ll |1-\phi/\phi_J(\varepsilon)|^\Delta$, a soft composite resides either in a matrix-dominated regime if $\phi < \phi_J$ or in a particle-dominant regime if $\phi > \phi_J$. As $G_m$ approaches zero, $G(\varepsilon, \phi) = G_m(1-\phi/\phi_J(\varepsilon))^{-\gamma}$ for $\phi < \phi_J$, or $G(\varepsilon, \phi) = \mathcal{C}G_p|1-\phi/\phi_J(\varepsilon)|^\beta$ for $\phi > \phi_J$, where $\mathcal{C}$ is a prefactor depending on the material parameters $c_1$ and $c_2$. When $1 \gg G_m/G_p \gg |1-\phi/\phi_J(\varepsilon)|^\Delta$, a soft composite is anticipated to be in the critical regime, where $G = c_1^{-\beta/\Delta}G_m^{\beta/\Delta}G_p^{1-\beta/\Delta}$.

## Discussion

The study reveals the essential role of shear-jamming in the mechanics of soft composites in the dense limit, a regime where the system becomes highly responsive and promises wide-ranging applications, yet remains challenging to model using conventional tools from continuous mechanics. We show that the strain-stiffening of soft composites can be interpreted as a manifestation of the criticality near a strain-dependent jamming point of dense suspensions (Fig. 5). The efficacy of our scaling model reveals the unique mechanical features of soft composites. As $G_m$ decreases to the order of $10^1$–$10^2$ Pa, the PDMS matrix consists of both a weakly crosslinked network and a substantial

amount of uncrosslinked free chains. The characteristic pore size of the network can be estimated as $a \sim (k_B T/G)^{1/3} \sim 50$ nm[47]. Therefore, the particles included in the PDMS matrix can potentially move to create direct contacts without causing fractures in the network. Consequently, the contact network within soft composites may resemble that in shear-jammed granular systems as $G_m$ approaches zero.

From the perspective of materials science, the study will benefit materials design in tissue engineering. Strain-stiffening has been widely observed in both biological[48] and synthetic tissues[10,16], with the prevailing interpretations attributing them to the nonlinear mechanics of the fibrous networks in the matrix. The significance of direct contacts between inclusions and the associated jamming transition in soft matrices began attracting attention only recently[16,49]. A key difference between our experiments and previous studies[10,48,49] is that the strain stiffening in our study occurs without increasing the volume fraction and thus cannot be explained by the model in ref. 16. The connection between strain-stiffening in incompressible soft composites and shear-jamming in dense suspensions offers a new scheme for designing the tissue-like mechanics of soft composites.

## Methods

### Material preparation

Both the PS and glass particles are micron-sized spheres with size distributions that can be described by the log-normal function $f(r) = \frac{1}{\sqrt{2\pi}\sigma r} \exp\left(-\frac{1}{2}\left(\frac{\ln(r/r_0)}{\sigma}\right)^2\right)$. For the PS particles, $r_0 = 12$ μm and $\sigma = 0.6$. For the glass particles, $r_0 = 20$ μm and $\sigma = 0.5$. The shear modulus of the particles, $G_p$, was measured by compressing individual beads between two flat substrates using a nanoindenter (Bruker, Hysitron TI-980). The resulting force-displacement curves were fitted to the Hertzian contact model (see Supplementary Fig. 1b). The results showed that $G_p = 1.6$ and 15.8 GPa for the PS and the glass particles, respectively.

The PDMS matrix was made by mixing a silicone base vinyl-terminated polydimethylsiloxane (DMS-V31, Gelest Inc) with copolymer crosslinkers (HMS-301, Gelest Inc) and a catalyst complex in xylene (SIP6831.2, Gelest Inc.). We prepared two mixture solutions, Gelest Part A and Gelest Part B, before curing. In particular, Part A consisted of a silicone base with 0.005 wt% catalyst, and Part B consisted of a silicone base with 10 wt% crosslinkers. By changing the weight ratio of A to B from 14.5:1 to 8:1, we varied $G_m$ from 0.04 to 4 kPa.

We prepared disk-shaped composite samples with 10 mm radius and 10 mm height in an acrylic mold covered with a para-film. To fully relax the internal structures, we used a vortex mixer (BV1000, Benchmark Scientific Inc.) to vibrate the samples immediately after mixing all the components. For $\phi > 0.5$, we compressed the samples using a glass plate to flatten the top surface. Each sample was then left to cure at room temperature for at least 48 h.

### Criticality analysis

We first show how the scaling form of the equations of the state shown in the main text Eq. (4) can be obtained by minimizing a scale-invariant phenomenological free energy. Denote the singular part of the free energy of a dense granular suspension ($G_m = 0$) under a given axial strain $\varepsilon$ as $F(\Phi, \mathcal{G})$, where $\mathcal{G} \equiv G/G_p$ is the dimensionless shear modulus, and $\Phi \equiv \phi/\phi_J(\varepsilon) - 1$ is the reduced volume fraction. For a given length scale $l$, we assume that the free energy is self-similar near the critical point $\Phi = 0$,

$$F(\Phi, \mathcal{G}) = l^{-d} F(l^{y_\Phi}\Phi, l^{y_\mathcal{G}}\mathcal{G}), \tag{9}$$

where $d = 3$ is the space dimension, and $y_\Phi$ and $y_\mathcal{G}$ are the scaling dimensions of $\Phi$ and $\mathcal{G}$, respectively. Considering $l = |\Phi|^{-\frac{1}{y_\Phi}}$, Eq. (9) can

be expressed as

$$F(\Phi, \mathcal{G}) = |\Phi|^{\frac{d}{y_\Phi}} \tilde{F}_\pm(|\Phi|^{-\frac{y_\mathcal{G}}{y_\Phi}}\mathcal{G}). \tag{10}$$

where $\tilde{F}_+$ and $\tilde{F}_-$ are different forms of the free energy in the regimes of $\Phi > 0$ and $\Phi < 0$, respectively.

For a composite with $G_m > 0$, the parameters $\{\mathcal{G}, G_m\}$ are analogous to $\{M, H\}$ in the Ising model. We define $\mathcal{G}_m \equiv G_m/G_p$ as the dimensionless shear modulus of the elastomer matrix. To transform the variable by substituting $\{\Phi, \mathcal{G}\}$ with $\{\Phi, \mathcal{G}_m\}$, we minimize the following Legendre transformation function:

$$\mathcal{L}(\Phi, \mathcal{G}_m) = \min_\mathcal{G}\{F(\Phi, \mathcal{G}) - \mathcal{G}_m\mathcal{G}\}. \tag{11}$$

In soft composites, $F(\Phi, \mathcal{G})$ and $\mathcal{L}(\Phi, \mathcal{G}_m)$ are in direct analogy to the Helmholtz free energy and Gibbs free energy in thermodynamic systems. The explicit evaluation of Eq. (11) leads to

$$\mathcal{G}_m = |\Phi|^\Delta \tilde{F}_\pm'(|\Phi|^{-\beta}\mathcal{G}), \tag{12}$$

where $\Delta \equiv (d - y_\mathcal{G})/y_\Phi$, and $\beta \equiv y_\mathcal{G}/y_\Phi$. By defining $f_\pm$ as the inverse functions of $\tilde{F}_\pm'$, we obtain the scaling form of the equations of state shown in Eq. (5) of the main text:

$$\mathcal{G} = |\Phi|^\beta f_\pm(\mathcal{G}_m|\Phi|^{-\Delta}). \tag{13}$$

For $\Phi < 0$, we have

$$\lim_{\mathcal{G}_m \to 0} \frac{\mathcal{G}}{\mathcal{G}_m} \sim \frac{\partial \mathcal{G}}{\partial \mathcal{G}_m} = |\Phi|^{\beta - \Delta} f_\pm'(|\Phi|^{-\Delta}\mathcal{G}_m) \sim |\Phi|^{\beta - \Delta}. \tag{14}$$

Compared with Eq. (2) in the main text, we have $\gamma = \Delta - \beta$.

In addition, Eq. (13) suggests that $f_\pm(\mathcal{G}_m|\Phi|^{-\Delta}) \propto (\mathcal{G}_m|\Phi|^{-\Delta})^{\beta/\Delta}$ at the critical point $\Phi = 0$ to prevent the divergence of free energy. Therefore, we have

$$\mathcal{G}(\Phi = 0) \sim \mathcal{G}_m^{\beta/\Delta}. \tag{15}$$

Compared with Eq. (3) in the main text, we obtain $\delta = \Delta/\beta$.

We next derive the explicit form of the equation of states in Eq. (5) in the main text. Based on the scale-invariant expression of Eq. (9), the expansion of $F(\Phi, \mathcal{G})$ should comprise terms $\Phi^a \mathcal{G}^b$ with $ay_\Phi + by_\mathcal{G} = d$. Therefore, $F$ can be expressed as

$$F(\Phi, \mathcal{G}) = \sum_i \mu_{i,\pm} |\Phi|^{a_i} \mathcal{G}^{\frac{d - a_i y_\Phi}{y_\mathcal{G}}} = \sum_i \mu_{i,\pm} |\Phi|^{a_i} \mathcal{G}^{\frac{\Delta - a_i}{\beta} + 1}, \tag{16}$$

where $a_i > 0$, and $\mu_{i,\pm}$ are the expansion coefficients for $\Phi > 0$ and $\Phi < 0$. By evaluating the variation in Eq. (11), we obtain

$$\mathcal{G}_m = \sum_i \mu_{i,\pm} \frac{\Delta - a_i + \beta}{\beta} |\Phi|^{a_i} \mathcal{G}^{\frac{\Delta - a_i}{\beta}}. \tag{17}$$

The above equation can be further simplified by including only three terms to describe the key experimental observations. First, $\lambda = \mu_0(\Delta + \beta)/\beta > 0$ when $a_0 = 0$ to ensure that the free energy is minimum at $\mathcal{G} = 0$ while $\Phi = 0$. Second, $\mu_{1,\pm}' = \mu_{1,\pm}(\Delta + \beta - 1)/\beta \neq 0$ when $a_1 = 1$ to ensure that $\partial \mathcal{G}(\Phi)/\partial \Phi \neq 0$ at $\Phi = 0$. Finally, because $\mathcal{G} \propto \mathcal{G}_m$ in the matrix-dominated regime, we have $\mu_{2,\pm}' = 2\mu_{2,\pm} \neq 0$ when $a_2 = \Delta - \beta$. As a consequence, $\mathcal{G}_m$ can be simplified as

$$\mathcal{G}_m = \lambda \mathcal{G}^{\frac{\Delta}{\beta}} + \mu_{1,\pm}' |\Phi| \mathcal{G}^{\frac{\Delta - 1}{\beta}} + \mu_{2,\pm}' |\Phi|^{\Delta - \beta} \mathcal{G}. \tag{18}$$

Due to the intrinsic nature of a continuous phase transition at $\Phi = 0$, we have $\mu_{i,\pm}' = \mp \mu_i'$ with $\mu_i' > 0$ for both $i = 1$ and 2. By defining

the reduced variables $\tilde{m} \equiv \mathcal{G}/|\Phi|^\beta$ and $\tilde{h} \equiv \mathcal{G}_m/|\Phi|^\Delta$, Eq. (18) can be rewritten as

$$\tilde{h} = c_1 \tilde{m}^{\frac{\Delta}{\beta}} \mp c_2 \tilde{m}^{\frac{\Delta-1}{\beta}} \mp c_3 \tilde{m}, \tag{19}$$

where $c_1 = \lambda$, $c_2 = \mu'_1$, and $c_3 = \mu'_2$. In the regime of $\Phi < 0$, we experimentally observed $\tilde{h}/\tilde{m} = 1$ as $\tilde{m} \to 0$, suggesting that $c_3 = 1$. Therefore, we finally obtain

$$\tilde{h} = c_1 \tilde{m}^{\frac{\Delta}{\beta}} \mp c_2 \tilde{m}^{\frac{\Delta-1}{\beta}} \mp \tilde{m}, \tag{20}$$

which is Eq. (5) in the main text.

In the particle-dominated regime, when $\Phi > 0$ and $\tilde{h} = 0$, the nonzero solution of $\tilde{m}$ from Eq. (20) gives the prefactor $\mathcal{C}$ in the scaling of the shear modulus $G = \mathcal{C} G_p |1 - \phi/\phi_J|^\beta$. The value $\mathcal{C}$ can be obtained by solving

$$c_1 \mathcal{C}^{\frac{\Delta}{\beta}-1} - c_2 \mathcal{C}^{\frac{\Delta-1}{\beta}-1} - 1 = 0, \tag{21}$$

and is thus determined by both $c_1$ and $c_2$.

In the critical regime, as $\Phi = 0$ and both $\tilde{m} \to \infty$ and $\tilde{h} \to \infty$, Eq. (20) reduces to $\tilde{h} = c_1 \tilde{m}^{\Delta/\beta}$, which gives $G = c_1^{-\beta/\Delta} G_m^{\beta/\Delta} G_p^{1-\beta/\Delta}$.

## Data availability
All the data supporting the findings of this study are available within the main text and the Supplementary Information. The source data used in generating all the figures are provided in the Source Data file. Source data are provided with this paper.

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

## Acknowledgements
We thank Bulbul Chakraborty, Yilong Han, Hisao Hayakawa, Ryohei Seto, and Xiang Cheng for the insightful discussions. The work was supported by the Early Career Scheme No. 26309620 (Q.X.), the General Research Fund No. 16307422 (Q.X.) and No. 16300221 (R.Z.), and the Collaborative Research Fund No. C6004-22Y (Q.X.) and No. C6008-20E (Q.X.) from the Hong Kong Research Grants Council (RGC). We also appreciate the support of the Partnership Seed Fund from the Asian Science and Technology Pioneering Institutes of Research and Education League No. ASPIRE2021#1 (Q.X. and Y.W.). Yiqiu Zhao acknowledges the support from the RGC postdoctoral fellowship PDFS2324-6S02 (Y.Z.). Hanqing Liu is supported by the U.S. Department of Energy, Office of Science, Nuclear Physics program, and by the Quantum Science Center.

## Author contributions
Y.Z., Y.W. and Q.X. designed the project. Y.Z., H.H., C.Y. and C.X. conducted the experimental measurements. Y.Z. and Q.X. analyzed the experimental data. H.L., Y.Z. and Q.X. built the scaling model. Y.H. and R.Z. performed the simulations on the isotropic jammed states. Y.Z., H.L. and Q.X. wrote the manuscript.

## Competing interests
The authors declare no competing interests.
