## [Peer Review File · Nature Communications]

REVIEWER COMMENTS

Reviewer #1 (Remarks to the Author):

Zhao et al. report their systematic experimental studies of the strain stiffening of soft composite solids with hard particles dispersing in soft matrices. They find that shear jamming has significant effects on the strain stiffening, beyond the scope of classical theories. In particular, they perform critical scalings near the critical volume fraction of the shear jamming transition and find that the same set of critical exponents apply to the shear jamming line extending from random close packing to shear jamming at high strain values.

Similar critical scalings have been used in previous studies to characterize the criticality of the jamming transition, for example, the seminal work by Olsson and Teitel (Ref. 18). As far as I know, the scaling of shear jamming has not been carefully studied. The authors have done pretty thorough and delicate work on this and come up with robust results. Moreover, they apply shear jamming to understand the non-classical behaviors of the soft composite solids, which would shed light on the design and manipulation of soft composite solids. The paper is well-written. The results and analysis are valid and convincing. I would like to recommend publication in Nature Communications after the authors address the following issues and make necessary changes to the manuscript.

(1) The PS-PDMS suspension should be non-Newtonian fluids, especially near the shear jamming transition. This can also be seen from Fig. S2. Is there a reason why the plateau viscosity is used in Eq. (1)? Is the PDMS base solution without crosslinkers Newtonian, so that the viscosity η_s is independent of applied shear stress?

(2) The authors show in Fig. S8 that the storage modulus exhibits a plateau when the frequency is below 0.1 rad/s for $G_m=0.35$ kPa. I think this is why they measure the shear modulus at 0.1 rad/s. Is it safe to apply the same frequency to much smaller values of G_m ? Intuitively, when G_m approaches 0, the low-frequency plateau of the storage modulus should decay and one needs to use smaller and smaller frequencies. This should be clarified in the paper.

(3) The critical scalings of $G_{r,max}(\phi_J-\phi)$ and $G_{max}(G_m)$ are crucial. Is there any possible explanation why the scaling exponent of $G_{max}(G_m)$ is $1/0.6$ and independent of $\phi_J(\epsilon)$ all the way to the random close packing? In the steady-state measurement of the viscosity, the shear strain ϵ is not relevant, so that η/η_s always diverges at the lowest ϕ_J below which shear jamming cannot occur. Why this scaling still applies to $G_{r,max}$ when ϕ_J varies?

(4) Minor points: Right below Eq. (5), ϕ_m is not defined when it first appears. In the caption of Fig. 1(d), solid and hollow points should indicate $G_{r,max}$ and $G_{r,0}$, respectively.

Reviewer #3 (Remarks to the Author):

The manuscript by Zhao et al. presents experimental studies on soft composite solids, specifically their shear responses as a function the volume fraction of inclusions. Previous efforts have failed to capture this response especially at high volume fraction. In a model soft composite solid with polystyrene

microspheres dispersed in polydimethylsiloxane elastomers, they have theoretically proposed and experimentally demonstrated that the strain-stiffening effect is governed by the critical behavior in the vicinity of shear-jamming transition. They further validate the proposed critical behavior in another soft composite solid.

The manuscript is well written, results are clearly presented and explained, most conclusions are well supported with data (see my comments below), and the authors were meticulous measuring all quantities. The findings are of great importance and the topic is of interest to the broad soft matter community. I will support the publication of the manuscript in Nature Communications if the following concerns are addressed.

1. I find it very interesting that the Poisson ratio of soft composite solids is 0.5, which serves as an excellent system to study shear. However, due to shear jamming, especially at large strains, I would expect shear induced compaction or dilation that are common in granular systems. As a result, this may lead to variation of the total volume of the solid. Can the authors comment on why that effect is suppressed or not present in the pure shear experiments shown in this work?

2. Can authors comment on why the rupture always occur approximate at a strain of 0.2, regardless of G_m ? Naively, one may expect that the susceptibility of PDMS to rupture is related to the crosslinking density.

3. It might be worth pointing out the condition to measure the viscosity of the PS-PDMS suspension in the main text, since the authors did observe shear thinning for a certain range of ϕ and τ .

4. I am confused about the value $\phi_J = 0.594$ for PS-PDMS composites. In my opinion, more data are needed to show that the behavior of η and G_{max} are caused by a shear jamming transition. If I understand this correctly, $\phi_J = 0.594$ is the lowest packing fraction needed in order to observe shear jamming, which presumably is below the isotropic jamming point for a frictional sphere packing (even though this value depends on the protocol to create the jammed packing). However, as stated in the Supplemental Materials IA2, $\phi_J = 0.599$ is interpreted as “the frictional jamming point of the PS particles”. Wouldn't this mean all composites with $\phi > \phi_J$ is already jammed even at $\epsilon = 0$?

5. Following up on comment 4, the PS-PDMS suspension already shows a non-zero shear modulus at $\phi > \phi_J$, doesn't that indicate that the suspension is already jammed? In addition, the non-zero shear modulus is measured at $\epsilon = 0.2$, more experiment protocol should be provided. Specifically, does the suspension still undergo pure shear if uniaxial compression is applied?

6. More on the value of ϕ_J . Various physical properties like friction and adhesion could affect the frictional jamming point. As mentioned in Supplemental Materials IA2, there may be weak attractions between PS particles. Previous works (e.g., Koeze and Tighe, Phys. Rev. Lett. 2018) have shown that the jamming point decreases significantly even with weak attractions between particles. In my opinion, it'll be worthwhile to perform another set of DEM simulations, where friction and/or adhesion are taken into consideration, to rule out the possibility that the behavior of η and G_{max} arises from the system already being jammed at $\phi > \phi_J$ regardless of ϵ .

7. Can authors comment on the exponents (β and Δ) observed in this study in comparison to previous studies on shear viscosity of granular materials, e.g., Ref. 18 (this is in 2D though) and Hatano, Journal of the Physical Society of Japan, 2008?

8. It is nice to see that Eq. 8 works for another soft composite solids when PS beads are replaced by glass beads. Can authors comment on the difference of ϕ_m and ϵ^* between PS and glass beads? Is the difference significant? And if so, what may cause the difference?

Response to the review report of NCOMMS-23-45685-T: *Elasticity-Controlled Jamming Criticality in Soft Composite Solids*

Dear Editors and Reviewers,

We are very grateful for the positive and thoughtful comments from both reviewers, which helped us to clarify many important issues in this work. Below, please find our point-by-point reply to all the comments from both reviewers, which are highlighted in blue. The revisions are marked in red in both the revised manuscript and the revised supplemental materials.

Reviewer #1 (Remarks to the Author):

Zhao et al. report their systematic experimental studies of the strain stiffening of soft composite solids with hard particles dispersing in soft matrices. They find that shear jamming has significant effects on the strain stiffening, beyond the scope of classical theories. In particular, they perform critical scalings near the critical volume fraction of the shear jamming transition and find that the same set of critical exponents apply to the shear jamming line extending from random close packing to shear jamming at high strain values.

Similar critical scalings have been used in previous studies to characterize the criticality of the jamming transition, for example, the seminal work by Olsson and Teitel (Ref. 18). As far as I know, the scaling of shear jamming has not been carefully studied. The authors have done pretty thorough and delicate work on this and come up with robust results. Moreover, they apply shear jamming to understand the non-classical behaviors of the soft composite solids, which would shed light on the design and manipulation of soft composite solids. The paper is well-written. The results and analysis are valid and convincing. I would like to recommend publication in Nature Communications after the authors address the following

issues and make necessary changes to the manuscript.

□ **Reply from the authors:**

We appreciate the positive comments from the reviewer. Below, we address the questions in detail.

Reviewer 1 comment 1

The PS-PDMS suspension should be non-Newtonian fluids, especially near the shear jamming transition. This can also be seen from Fig. S2. Is there a reason why the plateau viscosity is used in Eq. (1)? Is the PDMS base solution without crosslinkers Newtonian so that the viscosity η_s is independent of applied shear stress?

□ **Reply from the authors:**

We thank the reviewer for pointing out the important issues regarding the rheology of both PS-PDMS suspensions and PDMS base solutions. Here, we explain their rheological flow curves based on our measurements.

(1). The PDMS base solution without crosslinkers exhibits Newtonian fluid behaviors. Figure R1 represents the viscosities of base PDMS solution (η_s) measured at different shear stresses (τ). As τ increases from 10^{-1} Pa to 10^2 Pa, the viscosity remains constant, $\eta_s = 0.959 \pm 0.002$ Pa·s, indicating a characteristic Newtonian behavior.

FIG. R1. Plot of the viscosity of base PDMS against shear stress.

(2) We agree with the reviewer that the PS-PDMS suspensions can exhibit non-Newtonian flow behaviors with an increase in particle volume fraction (ϕ). As shown in Fig. S2(a) of Supplemental Materials, the PS-PDMS suspensions with $\phi \leq 0.5$ predominantly exhibit Newtonian rheology. However, for $\phi = 0.53$ and 0.55 , shear-thinning appears at $\tau < 1$ Pa, followed by a plateau regime between 1 and 10 Pa. **We argue that this plateau viscosity reflects the viscosity of suspensions in the quasi-static limit.**

To experimentally confirm this statement, Fig. R2 shows the plots of the relative viscosity η/η_s against the shear stress τ obtained under different measuring rates. The waiting time t_w is defined as the measuring duration at each shear stress, during which η/η_s is determined by averaging the measured viscosity of this period. As t_w increases from 100 s to 2 hrs, the flow curves gradually became more Newtonian-like at low stresses. However, mild shear thinning persists at $\tau < 1$ Pa even with $t_w = 2$ hrs. Given that the uncrosslinked PDMS polymers consist of long silicone chains (Molecular weight: $M_w = 2.8 \times 10^4$ g/mol), they relax very slowly in dense suspensions at low shear stresses (Xu, et al. Journal of Rheology, 64, 321 (2020) and Naald, et al. Soft Matter, 17, 3144 (2021)). We attribute the observed shear thinning behavior (for $\tau < 1$ Pa) to the transient rheology of the PS-PDMS suspensions that have not fully relaxed within t_w . Thus, the plateau viscosity (for $1 \text{ Pa} < \tau < 10 \text{ Pa}$) approximates the zero-shear viscosity in steady states. If t_w could be extended indefinitely, the viscosity below $\tau < 1$ Pa would be close to the plateau viscosity.

In addition, the strain-stiffening responses of dense composites were also characterized quasi-statically. We experimentally found that $\lim_{G_m \rightarrow 0}(G(\phi, \varepsilon = 0.2)/G_m) = \eta(\phi)/\eta_s = (1 - \phi/\phi_J)^{-2}$ (Eq. (2) in the main text), also supporting the suitability of using the plateau viscosity to determine ϕ_J .

To clarify the rheological properties of both the PDMS base solution and PS-PDMS suspensions, we have made the following changes in the revised manuscript:

(1) In the revised Fig. S2(a), we added the Newtonian flow curve of η/η_s against τ for the base PDMS solution. From line 81 to line 84 of revised Supplemental Materials,

FIG. R2. **[Revised Fig. S3]** Flow curves of the PS-PDMS suspension ($\phi = 0.55$) characterized with different measuring rates. For each shear stress τ , the relative viscosity η/η_s was determined by averaging the measured viscosity over a period of t_w . The waiting time t_w was varied systematically from 100 s to 2 hours.

we added the paragraph “Figure S2(a) shows the flow curves with varying particle volume fractions (ϕ), with the plot of $\phi = 0$ representing the Newtonian flow curve of the base PDMS”.

(2) We have included the Fig. R2 in the revised Supplemental Materials as Fig. S3.

(3) From line 93 to line 109 of the revised Supplemental Materials, we added the following paragraph “ We interpret the shear thinning observed for $\phi = 0.53$ and 0.55 below $\tau = 1$ Pa as a transient rheological response of dense PS-PDMS suspensions. To demonstrate this, Fig. S3 presents the flow curves of PS-PDMS suspensions having $\phi = 0.55$ characterized under different measuring rates. By defining t_w as the measuring duration at each τ , we observed a gradual transition of flow curves towards Newtonian-like behaviors as t_w increased from 100 s to 2 hours. Since the uncrosslinked PDMS solvents are made of longer polymer chains, their relaxation within dense suspensions can be very slow, particularly under low shear stresses. If the measuring period was extended significantly, we expected that the relative viscosity η/η_s below $\tau = 1$ Pa would align with that within the plateau regime, $1 \text{ Pa} < \tau < 10 \text{ Pa}$. Thus, the plateau

viscosities $\eta_{\text{plateau}}(\phi)$ were used to determine the shear-jamming volume fraction ϕ_J in Fig. S2(d)”.

Reviewer 1 comment 2

The authors show in Fig. S8 that the storage modulus exhibits a plateau when the frequency is below 0.1 rad/s for $G_m = 0.35$ kPa. I think this is why they measure the shear modulus at 0.1 rad/s. Is it safe to apply the same frequency to much smaller values of G_m ? Intuitively, when G_m approaches 0, the low-frequency plateau of the storage modulus should decay and one needs to use smaller and smaller frequencies. This should be clarified in the paper.

□ Reply from the authors:

We appreciate the insightful suggestions from the reviewer. In all of our rheological measurements, we observed that $\omega = 0.1$ rad/s is sufficiently small to reach the quasi-static plateau of the storage moduli. For example, in addition to $G_m = 0.35$ kPa, Fig. R3 also show the rheological properties of PS-PDMS composites composed of the softest matrix ($G_m = 0.04$ kPa), while keeping the volume fraction constant ($\phi = 0.57$). In line with the reviewer’s comment, we did observe a decrease in the threshold ω of the low-frequency plateau as G_m decreased, yet it consistently remained above $\omega = 0.1$ rad/s. Therefore, we conclude that the chosen measuring frequency ($\omega = 0.1$ rad/s) is appropriate for characterizing the composite moduli in this study.

To address this concern, we have made the following changes in the revised manuscript:

- (1) We have included Fig. R3 in the revised Supplemental Materials as the Fig. S11.
- (2) Between line 69 and line 71 of the revised main text, we added the following paragraph “ At an angular frequency ($\omega = 0.1$ rad/s), the storage modulus has reached a low-frequency plateau (Fig. S11), which indicates the shear modulus of soft composites (G). This ...”
- (3) Between line 271 to line 275 of the revised supplemental materials, we included the following paragraph “The shear moduli of dense composites were determined by G' measured at $\omega = 0.1$ rad/s. Figure S11 shows that $\omega = 0.1$ rad/s is sufficiently small to capture the low-frequency plateau of G' .”

FIG. R3. **[Revised Fig. S11]** Oscillatory rheology of PS-PDMS composites with a constant volume fraction ($\phi = 0.57$) and different matrix stiffnesses ($G_m = 0.35$ kPa and 0.04 kPa).

Reviewer 1 comment 3

The critical scalings of $G_{r,\max}(\phi_J - \phi)$ and $G_{\max}(G_m)$ are crucial. Is there any possible explanation why the scaling exponent of $G_{\max}(G_m)$ is $1/0.6$ and independent of $\phi_J(\varepsilon)$ all the way to the random close packing?

□ Reply from the authors:

We agree with the reviewer that the two scalings, $G_{r,\max}(\phi - \phi_J)$ (Eq. (2) in the main text) and $G_{\max}(G_m)$ (Eq. (3) in the main text), play a crucial role in our criticality framework. By systematically varying the axial strain (ε), we found that the scaling formula (Eq. (4)) agrees well with the experimental results measured from 150 composite samples with varying material parameters (Fig. 4). To reply the reviewer's comments, we would like to discuss the critical scalings the following from two perspectives.

First, we experimentally found that $G_{\max}(G_m) \sim G_m^{1/\delta}$ where $1/\delta = 0.6 \pm 0.1$. While the critical behaviors near shear jamming have been studied numerically (Ref. [40]), to our best knowledge, the scalings with respect to the media elasticity (e.g., $G_{\max} \sim G_m^{1/\delta}$) have never been investigated. In the case of connected fibrous networks, the shear modulus of networks (G_{network}) was found to follow a power-law scaling against the bending rigidity of the individual fibre (κ) using the lattice model: $G_{\text{network}} \sim \kappa^x$.

For a 2D triangular lattice, $x = 0.5$; for a 3D fcc lattice, $x = 0.4$. While these components are close to our measured index $1/\delta$, it remains unknown whether the indices from the two systems belong to the same universality class. A fundamental understanding of the origin of $1/\delta$ requires further investigations. This potential connection between soft composites and fibrous networks has already been mentioned between line 153 and line 155 of the main text.

Second, the previous simulation from Baity-Jesi *et al.* (Ref. [40]) has demonstrated that the scalings of orthogonal shear modulus against the excess contact number near shear-jamming transitions ($\varepsilon > 0$) is the same as that near isotropic-jamming transitions ($\varepsilon = 0$). The result suggests a constant symmetry along the shear jamming boundary. Thus, in the study of dense composites, we first assumed the strain-independent critical exponents and then experimentally verified the validity of this assumption.

To clarify this issue, we revised the paragraph

“ Motivated by the previous simulation showing the similar symmetry between shear-jamming and isotropic jamming transistons [40], we assume that the critical exponents ($\beta = 3$ and $\Delta = 5$) and the material parameters ($c_1 = 1.4$ and $c_2 = 1.3$) of dense composites remain constant for different ε . Therefore, Eq. 5 ...”

between line 177 to line 180 of the revised main text.

Reviewer 1 comment 4

Minor points: Right below Eq. (5), ϕ_m is not defined when it first appears. In the caption of Fig. 1(d), solid and hollow points should indicate $G_{r,\max}$ and $G_{r,0}$, respectively.

□ Reply from the authors:

We thank the reviewer for pointing out these typos.

(1) The ϕ_m below Eq. (5) should be ϕ_J . We have corrected the notations in line 166 and line 167 of the revised main text.

(2) We have also corrected the notations of $G_{r,\max}$ and $G_{r,0}$ in the revised caption of Fig.1.

Reviewer #3 (Remarks to the Author):

The manuscript by Zhao et al. presents experimental studies on soft composite solids, specifically their shear responses as a function the volume fraction of inclusions. Previous efforts have failed to capture this response especially at high volume fraction. In a model soft composite solid with polystyrene microspheres dispersed in polydimethylsiloxane elastomers, they have theoretically proposed and experimentally demonstrated that the strain-stiffening effect is governed by the critical behavior in the vicinity of shear-jamming transition. They further validate the proposed critical behavior in another soft composite solid.

The manuscript is well written, results are clearly presented and explained, most conclusions are well supported with data (see my comments below), and the authors were meticulous measuring all quantities. The findings are of great importance and the topic is of interest to the broad soft matter community. I will support the publication of the manuscript in Nature Communications if the following concerns are addressed.

Reply from the authors:

We thank the positive comments from the reviewer. Below, we address the posed questions in detail.

Reviewer 3 comment 1

I find it very interesting that the Poisson ratio of soft composite solids is 0.5, which serves as an excellent system to study shear. However, due to shear jamming, especially at large strains, I would expect shear induced compaction or dilation that are common in granular systems. As a result, this may lead to variation of the total volume of the solid. Can the authors comment on why that effect is suppressed or not present in the pure shear experiments shown in this work?

Reply from the authors:

We agree with the reviewer that the incompressibility (Poisson ration = 0.5) of soft composites is an essential material property that enables us to explore the role of shear-jamming in composite mechanics.

Unlike dry granular materials that can either dilate or contract under shear, dense composites are composed of densely packed particles in a soft PDMS matrix. The

crosslinked PDMS gels are incompressible, as mentioned in Ref. [26]. Consequently, soft composites are also incompressible, leading to a Poisson’s ratio of $\nu = 0.5$. Experimental confirmation of this is presented in Fig. S12 in the revised Supplemental Materials (or Fig. S9 in the original Supplemental Materials).

Between line 65 and line 66 of the revised main text, we added the sentence “Due to the incompressibility of crosslinked PDMS gels [26], ...”.

Reviewer 3 comment 2

Can authors comment on why the rupture always occurs approximately at a strain of 0.2, regardless of G_m ? Naively, one may expect that the susceptibility of PDMS to rupture is related to the crosslinking density.

□ Reply from the authors:

We thank the reviewer for raising this important issue. As shown in Fig. 1(c), our experimental findings indicate the plasticity onset $\varepsilon \approx 0.2$ independent of G_m . Near this critical strain, we frequently detected the emergence of fractures at the interface between the PS particles and the PDMS matrix. In fact, it is well known that the strength of a soft composite material commonly relies on the binding energy between the two phases (D. Hull and T. W. Clyne, *An Introduction to Composite Materials*, Cambridge University Press). As these interfacial fractures propagate beyond $\varepsilon > 0.2$, the axial compressions result in irreversible deformations (Fig. 1(c)).

The fracture toughness between the two phases is fundamentally related to the interfacial adhesion energy. A previous study of soft adhesion has shown that the adhesion energy of soft gels remains approximately independent of G_m (Ref. [28]). Consequently, it is expected that the plasticity onset of soft composites is also independent of G_m .

To clarify this issue, we added the following sentence “Since the adhesion energy at gel interfaces is approximately independent of the crosslinking density [28], the plasticity onset ($\varepsilon \approx 0.2$) remains nearly unchanged for various G_m ...” between line 85 and line 87 in the revised main text.

Reviewer 3 comment 3

It might be worth pointing out the condition to measure the viscosity of the PS-PDMS

suspension in the main text, since the authors did observe shear thinning for a certain range of ϕ and τ .

□ Reply from the authors:

We thank the reviewer for the suggestion. In our experiments, we selected the plateau viscosity within the range of $1 \text{ Pa} < \tau < 10 \text{ Pa}$, as indicated in the revised Fig. S2(a), to determine ϕ_J in Fig. 2(a). This measurement protocol can be justified from the following two perspectives:

FIG. R4. **[Revised Fig. S3].** Flow curves of the PS-PDMS suspension ($\phi = 0.55$) characterized with different measuring rates. For each shear stress τ , the relative viscosity η/η_s was determined by averaging the measured viscosity over a period of t_w . The waiting time t_w was varied systematically from 100 s to 2 hours.

First, the plateau viscosity measured between 1 Pa and 10 Pa effectively captures the viscosity in the quasi-static limit. Figure R4 (same as Fig. R2) shows the relative viscosity η/η_s against shear stress τ measured with different measuring rates. We define the waiting time t_w as the measuring duration at each shear stress. As t_w increases from 100 s to 2 hrs, the flow curves gradually became more Newtonian-like at low shear stresses. Even for $t_w = 2$ hrs, a mild shear thinning still appeared at $\tau < 1 \text{ Pa}$. Since the uncrosslinked PDMS are made of long silicone chains

(MW: $M_w = 2.8 \times 10^4$ g/mol), they relax very slowly in dense suspensions at low shear stresses (Xu, et al. *Journal of Rheology*, 64, 321 (2020) and Naald, et al. *Soft Matter*, 17, 3144 (2021)). We interpret the shear thinning at $\tau < 1$ Pa as the transient rheology of the PS-PDMS suspensions that were not fully relaxed within t_w . If t_w could be extended indefinitely, we anticipate that the suspension viscosity below $\tau < 1$ Pa would represent the plateau viscosity within the range of $1 \text{ Pa} < \tau < 10 \text{ Pa}$.

Second, the plateau viscosity effectively determines ϕ_J , which establishes the inherent connection between suspension rheology and composite mechanics. Using the plateau viscosity $\eta(\phi)$ between 1 Pa and 10 Pa, we experimentally confirmed that $\lim_{G_m \rightarrow 0} (G(\phi, \varepsilon = 0.2)/G_m) = \eta(\phi)/\eta_s = (1 - \phi/\phi_J)^{-2}$ where $\phi_J = 0.594$ (refer to Fig. 2(b) and Eq. (2) in the main text) for $\phi < \phi_J$. For $\phi > \phi_J$, we quantitatively measured a finite rigidity of the suspensions under $\varepsilon = 0.2$ (refer to Fig. 2(a)). The experimental evidence in both regimes further supports the appropriateness of using the plateau viscosity to determine ϕ_J .

To clarify the issue on our rheological measurements, we made the following revisions:

- (1) We have incorporated Fig. R4 to the revised Supplemental Materials as **Fig. S3**.
- (2) From line 93 to line 109 of the revised Supplemental Materials, we added the following paragraph “ We interpret the shear thinning observed for $\phi = 0.53$ and 0.55 below $\tau = 1$ Pa as a transient rheological response of dense PS-PDMS suspensions. To demonstrate this, Fig. S3 presents the flow curves of PS-PDMS suspensions having $\phi = 0.55$ characterized under different measuring rates. By defining t_w as the measuring duration at each τ , we observed a gradual transition of flow curves towards Newtonian-like behaviors as t_w increased from 100 s to 2 hours. Since the uncrosslinked PDMS solvents are made of longer polymer chains, their relaxation within dense suspensions can be very slow, particularly under low shear stresses. If the measuring period was extended significantly, we expected that the relative viscosity η/η_s below $\tau = 1$ Pa would align with that within the plateau regime, $1 \text{ Pa} < \tau < 10 \text{ Pa}$. Thus, the plateau viscosities $\eta_{\text{plateau}}(\phi)$ were used to determine the shear-jamming volume fraction ϕ_J in Fig. S2(d)”.
- (3) We added the following sentence “The left panel in Fig. 2(a) shows η_r measured within a Newtonian regime where the shear stress ranges from 1 Pa to 10 Pa. This

value effectively estimates the suspension viscosity in the quasi-static limit. The details of the rheological measurements are provided in Fig. S2.” between line 109 and line 112 of the revised main text.

Reviewer 3 comment 4

I am confused about the value $\phi_J = 0.594$ for PS-PDMS composites. In my opinion, more data are needed to show that the behavior of η and G_{\max} are caused by a shear jamming transition. If I understand this correctly, $\phi_J = 0.594$ is the lowest packing fraction needed in order to observe shear jamming, which presumably is below the isotropic jamming point for a frictional sphere packing (even though this value depends on the protocol to create the jammed packing). However, as stated in the Supplemental Materials IA2, $\phi_J = 0.599$ is interpreted as “the frictional jamming point of the PS particles”. Wouldn’t this mean all composites with $\phi > \phi_J$ is already jammed even at $\varepsilon = 0$?

□ Reply from the authors:

We thank the reviewer for asking this question, which provides an opportunity for us to further clarify the physical meaning of ϕ_J . Considering the inherent measuring uncertainties, $\phi_J = 0.594$ obtained from the rheology of PS-PDMS suspensions aligns well with $\phi_J = 0.599$ measured from PS-glycerol suspensions (Fig. S2). **The value of ϕ_J indicates the lowest particle volume fraction required to observe shear jamming.** We agree with the referee that the phrase “the frictional jamming point of the PS particles” used in the original Supplemental Materials may not be entirely clear. In the revised Supplemental Materials (from line 121 to line 128), we included the following paragraph:

“The PS-water/glycerol suspensions exhibited classical shear thickening behaviors. Considering the shear thickening mechanism in the suspension rheology [4], we interpret $\phi_J = 0.599$ as the minimum particle volume fraction needed to observe frictional shear jamming in PS-water/glycerol suspensions. As this ϕ_J is consistent with that obtained from PS-PDMS suspensions, we expect that the shear-jamming of the PS particles in PDMS is also governed by frictional contacts. ”

There are three experimental results demonstrating that our samples were not jammed at $\phi > \phi_J$. First, we have shown in Fig. R5(e) (or the original Fig.

FIG. R5. **[Revised Fig. S4] Strain-induced rigidity transition in PS-PDMS suspensions.**

(a) Schematic of the **experimental protocol to measure shear-jamming transitions in PS-PDMS suspensions**. The parallel-plate was initially filled up with a suspension under a gap size $d_0 = 1.5$ mm (state 1). The top plate was then lifted up to a new position with $d_1 > d_0$ (state 2). At this position, the suspension was fully relaxed due to an oscillatory shear with an angular frequency $\omega = 10$ rad/s and a strain amplitude $\delta\gamma_a = 10\%$ for 30 mins (state 3). State 3 was considered as the reference state ($\varepsilon = 0$). Finally, the top plate was lowered again to the initial position with a gap size d_0 (state 4), and the resulting axial strain is $\varepsilon = (d_1 - d_0)/d_1$. (b) Time evolution of the storage modulus (G') and loss modulus (G'') measured in state 4 with $d_1 = 1.76$ mm. An angular frequency of $\omega = 0.1$ rad/s and a strain amplitude of $\delta\gamma_a = 0.01\%$ were applied during the oscillatory tests. The plateau values of G' are denoted as the shear modulus of the jammed suspensions (G). (c) Time evolution of the normal stress (σ_{zz}) measured in state 4. (d – e) Normal stress σ_{zz} and shear modulus G of a suspension with $\phi = 0.62$ as the function of ε (**blue circles**). Both σ_{zz} and G begin to become nonzero near $\varepsilon_J = 0.05$, **indicating a shear jamming transition**. The grey stars in (e) show the strain-dependent shear modulus of a PS-PDMS composite sample with $\phi = 0.62$ and $G_m = 1.28$ kPa. (f – g) Normal stress σ_{zz} and shear modulus G of suspensions measured under $\varepsilon = 0$ and $\varepsilon = 0.2$, respectively, for samples with $\phi > \phi_J$. The black arrow shows the shear jamming process for $\phi = 0.62$ induced by an axial strain.

S3(e)) that the shear modulus of a dense PS-PDMS suspensions with $\phi = 0.62 > \phi_J$ is zero at $\varepsilon = 0$. Additionally, the shear moduli of the PS-PDMS suspensions measured at

FIG. R6. **[New Fig. S5] Liquid-to-solid transition of PS-PDMS suspensions ($\phi = 0.61$) induced by shear.** (a) Plots of the accumulative shear strain γ against time t under constant shear stresses, $\tau = 0.5$ Pa and 5.0 Pa, respectively. (b) Plots of shear rate $\dot{\gamma}$ against shear strain γ under the two different shear stresses.

different axial strains (blue circles) are close to those of a cured PS-PDMS composites with the same particle volume fraction $\phi = 0.62$ (grey stars). This agreement suggests that the dense composite is also unjammed at $\varepsilon = 0$.

Second, we prepared PS-PDMS suspension samples with three different volume fractions $\phi = 0.61$, 0.62, and 0.63. Each sample was fully relaxed through an oscillatory preshear, which was described between line 143 and line 148 of the Supplemental Materials. The state of $\varepsilon = 0$ represents the fully relaxed suspensions. Figures R5(f) and (g) show that the normal stresses σ_{zz} and the shear moduli G remain zero for all the samples at $\varepsilon = 0$, indicating that the samples were unjammed initially. As ε increased to 0.2, we observed a finite increase in both σ_{zz} and G for all the samples, which were experimental signatures of shear-jammed states.

Third, inspired by the simulation from Seto et al. (Granular Matter, 21, 82 (2019), now added as Ref. [43]), we applied a constant shear stress (τ) to an initially unjammed PS-PDMS suspension at $\varepsilon = 0$ ($\phi = 0.61$). As shown in Fig. R6, the shear rates ($\dot{\gamma}$) are initially non-zero and then abruptly drop to zero at a finite strain. This liquid-to-solid transition experimentally indicates the shear jamming transition of the suspensions. The non-zero shear-rate at small strains verify that the initial states are unjammed.

To better clarify that ϕ_J represents the lowest volume fraction for shear jamming

instead of isotropic jamming point, we made the following revisions:

(1) We included the revised Fig. R5 and new Fig. R6 as the Fig. S4 and Fig. S5, respectively, in the revised Supplemental Materials.

(2) Between line 116 and line 121 in the main text, we included the following paragraph to clarify the measurement protocol: “Instead, the suspensions were consistently jammed under continuous shear (Fig. S5). To quantify the mechanical responses of these shear-jammed states, we initially prepared fully relaxed suspensions without rigidity at $\varepsilon = 0$. Subsequently, we applied an axial strain $\varepsilon > 0$ to induce jamming in the suspensions and measure their shear moduli. The measurement protocol is detailed in Sec. I. A. 3 of the Supplemental Materials. The right panel of Fig. 2(a) represents the ...”.

(3) Between line 164 and line 179 of the revised Supplemental Materials, we added the following paragraph “Figure S4(e) shows the shear modulus ($G(\varepsilon)$) of a cured PS-PDMS composite with $\phi = 0.62$ and $G_m = 1.28$ kPa (grey stars), closely matching that of an uncured PS-PDMS suspension with the same particle volume fraction (blue circles).

To further demonstrate that the PS-PDMS suspensions were unjammed at $\varepsilon = 0$, Figs. S4(f) and (g) show that the normal stresses (σ_{zz}) and shear moduli (G) of the suspensions with $\phi = 0.61, 0.62$, and $0.63 (> \phi_J)$ rise from zero to finite values, respectively, as ε increases from 0 to 0.2. Additionally, Fig. S5 demonstrates that the suspension with $\phi = 0.61 > \phi_J$ can undergo a jamming transition under a simple shear with constant shear stresses. The regimes with non-zero shear rates ($\dot{\gamma}$) in Fig. S5 suggest that the PS-PDMS suspensions with $\phi = 0.61$ were initially unjammed.”

to interpret the results in both Figs. S4 and S5.

(4) We have changed the legend of Fig. 2(a) from “shear modulus” to “shear modulus ($\varepsilon = 0.2$)”, and have made revisions to the caption of Fig. 2 to better describe the rheological measurements of PS-PDMS suspensions:

“ In the regime of $\phi > \phi_J$, the suspensions were initially unjammed at $\varepsilon = 0$, and then shear jammed at $\varepsilon = 0.2$. The red crosses represent their shear moduli (G_s) in the shear-jammed states ($\varepsilon = 0.2$).”

Reviewer 3 comment 5

Following up on comment 4, the PS-PDMS suspension already shows a non-zero shear modulus at $\phi > \phi_J$, doesn't that indicate that the suspension is already jammed? In addition, the non-zero shear modulus is measured at $\varepsilon = 0.2$, more experiment protocol should be provided. Specifically, does the suspension still undergo pure shear if uniaxial compression is applied?

□ Reply from the authors:

First, in the reply to comment 4, we have demonstrated that the PS-PDMS suspensions with $\phi > \phi_J$ were not jammed at $\varepsilon = 0$, but can be shear-jammed by increasing the axial strains. The measurement protocol to quantify the shear modulus has been added to both the revised main text (from line 116 to line 121) and the revised Supplemental Materials (from line 164 to line 179).

Second, due to the incompressibility of uncrosslinked PDMS polymers, the PS-PDMS suspensions are also incompressible. **Similar to the PS-PDMS composites, the PS-PDMS suspensions also undergo pure shear when subjected to uni-axial compressions.**

Figure R7(a) shows a snapshot of the video illustrating a PS-PDMS suspension with $\phi = 0.61$ under strain in a parallel-plate cell. While applying axial compressions to the suspensions, we simultaneously capture the suspension boundaries using a high-resolution digital camera. We denote $x_{\text{left}}(y)$ and $x_{\text{right}}(y)$ as the traces of the suspension-air interfaces on both sides. Assuming azimuthal symmetry, the volume can be calculated as

$$V = \int_{y_0}^{y_1} \pi \left(\frac{x_{\text{right}}(y) - x_{\text{left}}(y)}{2} \right)^2 dy \quad (\text{R1})$$

where y_0 and y_1 are the positions of the bottom and top plates, respectively. Figures R7(b) and (c) demonstrate that the volume of the suspension remains unchanged under axial strains.

To show the incompressibility of PS-PDMS suspensions, we have included the Supplemental Video 1, with a snapshot depicted in Fig. R7. In the revised Supplemental

FIG. R7. [Snapshot of Supplementary Video 1]

Materials (from line 300 to line 303), we have added the following sentence “Similarly, we have experimentally verified that the volume of an uncured PS-PDMS suspension remains conserved under axial strains (see *Supplementary Video 1*)”.

Reviewer 3 comment 6

More on the value of ϕ_J . Various physical properties like friction and adhesion could affect the frictional jamming point. As mentioned in Supplemental Materials IA2, there may be weak attractions between PS particles. Previous works (e.g., Koeze and Tighe, Phys. Rev. Lett. 2018) have shown that the jamming point decreases significantly even with weak attractions between particles. In my opinion, it’ll be worthwhile to perform another set of DEM simulations, where friction and/or adhesion are taken into consideration, to rule out the possibility that the behavior of η and G_{\max} arises from the system already being jammed at $\phi > \phi_J$ regardless of ε .

□ Reply from the authors:

We agree with the reviewer that jamming fractions depend on particle interactions. However, the suspensions are not jammed at $\phi > \phi_J$ regardless of ε . In the reply to comment 4, we have shown three experiments to demonstrate that the PS-PDMS suspensions with $\phi > \phi_J$ were not jammed at $\varepsilon = 0$. With the increase of ε , the suspen-

sions underwent shear jamming at critical non-zero strains. The revisions addressing this issue have been summarized in the reply to comment 4.

7. Can authors comment on the exponents (β and Δ) observed in this study in comparison to previous studies on shear viscosity of granular materials, e.g., Ref. 18 (this is in 2D though) and Hatano, Journal of the Physical Society of Japan, 2008?

□ Reply from the authors:

We thank the reviewer for pointing out these two papers. Below, we compare the scaling exponents measured in our study with those documented in the two papers. To avoid any potential ambiguity, our discussion will adhere consistently to the notations of exponents (β , Δ , and γ) employed in our study.

Equation 2 of the main text is written as

$$\lim_{G_m \rightarrow} \frac{G_{\max}(\phi)}{G_m} = \frac{\eta(\phi)}{\eta_s} = (1 - \phi/\phi_J)^{-\gamma}, \quad (\phi < \phi_J) \quad (\text{R2})$$

where $\gamma = 2$. This value was determined to be 1.7 in Ref. [18] (denoted as β in the paper). Similarly, the value for an over-damped 3D system was also found to be 1.7 in Hatano’s paper. Given the complexity of particle interactions, we conclude that our empirical scaling of Eq. 2 is consistent with the outcomes of their simulations.

Equation (3) of the main text gives the second scaling relationship

$$G_{\max} \sim G_m^{1/\delta}, \quad (\phi = \phi_J) \quad (\text{R3})$$

where $1/\delta = \beta/\Delta = 0.6$ and G_m is the matrix modulus. Since both Ref. [18] and Hatano’s paper focus on granular systems without an elastic matrix, the scaling of Eq. (3) is not addressed in these works.

In the revised main text (from line 114 to line 115), we added the sentence “**A similar scaling has been identified in the simulations of over-damped granular systems near jamming [18, 31].**”

8. It is nice to see that Eq. 8 works for another soft composite solids when PS beads are replaced by glass beads. Can authors comment on the difference of ϕ_m and ε^* between PS and glass beads? Is the difference significant? And if so, what may cause the difference?

□ Reply from the authors:

Based on the results shown in Fig. 4(c), we conclude that ϕ_m of the PS beads (0.594 ± 0.002) differs from that of glass beads (0.613 ± 0.003). As ϕ_m determines the frictional jamming points (Fig. S2), this variance may stem from the different frictional coefficients and polydispersities between PS and glass beads.

However, given the experimental uncertainties, the difference in ε^* between PS beads (0.035 ± 0.003) and glass beads (0.040 ± 0.007) is insignificant. We notice that there was a typo in the old main text: for glass beads, $\varepsilon^* = 0.040 \pm 0.007$ instead of 0.040 ± 0.07 . This typo has been fixed in the revised manuscript.

In the revised main text (from line 203 to line 205, the following sentence has been added: “The variation in ϕ_m may stem from the difference in frictional coefficients and polydispersities between the PS and glass particles. ”

REVIEWERS' COMMENTS

Reviewer #1 (Remarks to the Author):

The authors have addressed my concerns. I am glad to recommend publication in Nature Communications as is.

Reviewer #3 (Remarks to the Author):

All my comments have been addressed. I now support the publication of the manuscript.

Response to the review report of NCOMMS-23-45685: *Elasticity-Controlled Jamming Criticality in Soft Composite Solids*

Dear Editors and Reviewers,

We are very grateful for the time and effort put in by both reviewers. As both reviewers have recommended the publication of our paper, we revised the manuscript based on the editorial requests. We believe that the revised manuscript meets the standards for publication in *Nature Communications*.

Reviewer #1 (Remarks to the Author):

The authors have addressed my concerns. I am glad to recommend publication in Nature Communications as is.

Reply from the authors: We thank the reviewer for the recommendation for publication.

Reviewer #3 (Remarks to the Author):

All my comments have been addressed. I now support the publication of the manuscript.

Reply from the authors: We thank the reviewer for the recommendation for publication.